# Daily Intake of Fermented Milk Containing *Lactobacillus casei* Shirota (Lcs) Modulates Systemic and Upper Airways Immune/Inflammatory Responses in Marathon Runners

**DOI:** 10.3390/nu11071678

**Published:** 2019-07-22

**Authors:** Mauro Vaisberg, Vitoria Paixão, Ewin B. Almeida, Juliana M. B. Santos, Roberta Foster, Marcelo Rossi, Tania C. Pithon-Curi, Renata Gorjão, Cesar M. Momesso, Marília S. Andrade, José R. Araujo, Maurício C. Garcia, Moises Cohen, Elizabeth C. Perez, Alana Santos-Dias, Rodolfo P. Vieira, André L. L. Bachi

**Affiliations:** 1Department of Otorhinolaryngology, Federal University of São Paulo (UNIFESP), São Paulo-SP 04039-032, Brazil; 2Institute of Physical Activity Science and Sport, Cruzeiro of Sul University, São Paulo-SP 01506-000, Brazil; 3Department of Physiology, Federal University of São Paulo (UNIFESP), São Paulo-SP 04023-062, Brazil; 4Department of Morphology, Federal University of São Paulo (UNIFESP), São Paulo-SP 04023-900, Brazil; 5Department of Orthopaedics and Traumatology, Federal University of São Paulo (UNIFESP), São Paulo-SP 04024-002, Brazil; 6Environmental and Experimental Pathology, Paulista University (UNIP), São Paulo-SP 04043-200, Brazil; 7Children´s Institute, Faculty of Medicine, University of São Paulo, São Paulo-SP 05403-000, Brazil; 8Post-graduation Program in Bioengineering and in Biomedical Engineering, Universidade Brasil, São Paulo-SP 08230-030, Brazil; 9Post-graduation Program in Sciences of Human Movement and Rehabilitation, Federal University of São Paulo (UNIFESP), Santos-SP 11060-001, Brazil; 10Brazilian Institute of Teaching and Research in Pulmonary and Exercise Immunology (IBEPIPE), São José Dos Campos-SP 12245-520, Brazil; 11School of Medicine, Anhembi Morumbi University, São José dos Campos-SP 12230-002, Brazil

**Keywords:** probiotics, cytokines, secretory immunoglobulin A, antimicrobial peptides, neutrophil infiltration

## Abstract

Background. Although *Lactobacillus casei* Shirota (LcS) can benefit the immune status, the effects of LcS in the immune/inflammatory responses of marathon runners has never been evaluated. Therefore, here we evaluated the effect of daily ingestion of fermented milk containing or not LcS in the systemic and upper airway immune/inflammatory responses before and after a marathon. Methods. Forty-two male marathon runners ingested a fermented milk containing 40 billion of LcS/day (LcS group, *n* = 20) or placebo (unfermented milk, *n* = 22) during 30 days pre-marathon. Immune/inflammatory parameters in nasal mucosa and serum, as well as concentrations of secretory IgA (SIgA) and antimicrobial peptides in saliva, were evaluated before and after fermented milk ingestion, immediately, 72 h, and 14 d post-marathon. Results. Higher proinflammatory cytokine levels in serum and nasal mucosa, and also lower salivary levels of SIgA and antimicrobial peptides, were found immediately post-marathon in the placebo group compared to other time points and to LcS group. In opposite, higher anti-inflammatory levels and reduced neutrophil infiltration on nasal mucosa were found in the LcS group compared to other time points and to the placebo group. Conclusion. For the first time, it is shown that LcS is able to modulate the systemic and airways immune responses post-marathon.

## 1. Introduction

One of the most relevant aspects related to the interaction between exercise and immune response is that athletes undergoing high-intensity efforts show increased incidence of upper respiratory tract infections (URTI), both in the context of competitions and during strenuous training [1]. Although former studies have suggested that the mechanism causing the manifestations of upper airways were usually related to infections, it has also been demonstrated that both inflammatory and allergic events in airway may also be responsible by such alterations. Therefore, the study of immune response involved in the protective mechanisms of the upper airways, including immune cells, antimicrobial peptides, immunoglobulins, and cytokines are essential to elucidate the mechanisms responsible for the breakdown of homeostasis represented by infection, inflammation, or allergic symptoms, all associated with activation of the inflammatory response [2,3]. 

Corroborating this idea, alterations in IL-6 and IL-10 concentrations both in serum and in the nasal mucosa were found in marathon runners who presented upper airways symptoms compared to the marathon runners who did not present symptoms [4]. The authors highlighted that the maintenance of a non-inflammatory environment in the upper respiratory tract can avoid the development of upper airways symptoms in marathon runners.

Lactobacilli, characterized as nonpathogenic lactic acid bacteria that are resistant to gastric acid and bile, are recognized as probiotics, which can be defined as live bacteria with a beneficial effect on the host when administered in adequate amounts. For more than 70 years, the food industry has used this type of bacteria to develop different products, mainly aimed to protect and recover the intestinal function [5,6]. Another beneficial characteristic of lactobacilli on the individuals´ health is the ability to regulate mucosal and systemic immune system. Although the effects of probiotics in human health have been extensively studied in several contexts, fewer studies have evaluated the role of probiotics in athletes. It was demonstrated that the ingestion of strains of lactobacilli (*Lactobacillus rhamnosus* [7] and *Lactobacillus casei* Shirota [8]) had protective effects on the incidence of respiratory and gastrointestinal tract symptoms during the training period and two weeks after a marathon. Moreover, the frequency, duration, and severity of these symptoms were decreased in lactobacilli supplemented individuals [7,8]. However, further in vivo studies are needed to clarify the effects and mechanisms involved in the health of athletes in use of probiotics, especially on issues such as the dose and duration of supplementation. 

Therefore, the present study aimed to evaluate the influence of the daily intake of *Lactobacillus casei* Shirota (LcS) on the systemic and upper airway immune/inflammatory responses before and after a race in marathon runners who previously reported upper respiratory symptoms (URS) after an exhaustive physical exercise session.

## 2. Material and Methods

### 2.1. Subjects and Study Design

According to the flow diagram (Figure 1), forty-two male amateur marathon runners (aged 39.5 ± 9.4), living in the São Paulo city and surrounding areas, were enrolled in the study after they signed the free assent consent term, previously approved by the Ethics Committee of the Cruzeiro do Sul University and the National Research Ethics Committee (CONEP) under number CAAE: 67318317.3.0000.8084. All experiments were performed at Institute of Physical Activity Science and Sport belonged to Cruzeiro of Sul University and at Department of Otorhinolaryngology belonged to Federal University of São Paulo (UNIFESP). The study was in agreement with the Ethical Standards defined by Harriss and Atkinson [9] and also with the Declaration of Helsinki.

Our inclusion criteria were the previous occurrence of URS after an exhaustive exercise session and the participation in the marathon race. After that, all volunteers were submitted to clinical examinations and cardiopulmonary testing in order to characterize their cardiorespiratory capacity and to body composition analysis by dual X-ray absorptiometry (DXA) (GE Healthcare Lunar Madison, WI, USA). These data were used together with the last race record to perform groups´ stratification.

The volunteers were instructed to keep their usual training/physical exercise schedules and also to report the occurrence and duration of upper respiratory symptoms (URS) or gastrointestinal disturbances throughout the study. The last training session was performed 24h before sample collection, except for the occasion named “immediately after the marathon”.

### 2.2. Schedule of Fermented Milk Ingestion

After group stratification, volunteers were separated into two groups: (1) LcS group (*n* = 20), which ingested 1 bottle of fermented milk (80 grams) with 40 × 10^9^ live cells of *Lactobacillus casei* strain Shirota per day and (2) placebo group (*n* = 22), which ingested 1 bottle of unfermented milk (80 grams) without the bacteria presence. Both groups began the intake period 30 days before the marathon competition. All the volunteers were oriented to interrupt this ingestion immediately and seek a medical attention if any gastrointestinal disturbance were perceived. To guarantee the development of a double-blind study, all bottles used in the study were kindly provided by Yakult S/A Indústria e Comércio, Brasil. These bottles were named with letters A and B by the manufacturer and presented the same taste, color, flavor, and pH. Information about the presence or absence of LcS in the bottles was provided only after the end of the study and all analysis were ready from the manufacturer.

### 2.3. Collection of the Samples

Fasting blood, saliva, cells, and lavage from the nasal mucosa, which can represent the upper airways tract, were collected at five different occasions: before (Pre) and 30 days after ingestion of fermented milk or placebo, which also corresponded to the 24 hours before the competition (Post-ingestion); immediately; 72 hours; and 14 days after the competition (Figure 2).

Blood samples were collected from a peripheral vein and, after blood clotting, tubes were centrifuged at 900× *g* for 10 min at 4 °C. A minimum of 1000 μL of serum was stored at −80 °C for later use to determine cytokine concentration.

Two milliliters of the whole saliva sample from each volunteer were collected directly into sterile 15 mL Falcon^®^ tubes without prior stimulation [4]. After collection, each sample was centrifuged at 3000 rpm for 10 min at 4 °C and 500 μL of supernatant were transferred to 1.5 mL Eppendorf tubes and kept frozen at −80 °C for later use to measure SIgA and antibacterial peptides concentration. No buffers or preservatives were added.

To determine the percentage of neutrophils infiltrating in the nasal mucosa, a swab was introduced into each nostril of the volunteers and the cells obtained were placed in a slide, fixed with methanol (99%, Synth), and stained with hematoxylin. An experienced pathologist analyzed all the slides.

After the procedure described above, samples of nasal mucosa lavage were obtained by introducing 10 mL of phosphate buffered saline (PBS-1x) into each nostril of the volunteer using a needleless syringe. The volunteer was instructed to remain with the liquid inside in the upper airways for a period of 10 s and after that, return by the nostril as much liquid as possible in a Falcon tube of 50 mL. The collected material was kept refrigerated at 4 ºC for 15 min. Subsequently, the total volume collected was measured using a graduated test tube and centrifuged at 900× *g* for 10 min at 4 ºC. A volume of 1500 L of the supernatant were transferred to 2 mL eppendorf´s tubes and stored at −80 °C for later use to determine cytokine concentration.

### 2.4. Determination of the Salivary Concentration of Immunoglobulin A (SIgA) and Antibacterial Peptides 

Salivary concentration of SIgA and antibacterial peptides LL-37 (cathelicidin), defensin- 1, lactoferrin, and lysozyme was determined by ELISA test using the Salivary Secretory IgA Indirect Enzyme Immunoassay kit (Bioassay Technology Laboratory, Shangai, China) according to the manufacturer’s instructions.

### 2.5. Determination of Cytokine Concentrations in Nasal Mucosa Lavage

Cytokines concentration (IL-1β, IL-1ra, IL-4, IL-5, IL-6, IL-10, IL-12p70, IL-13, and TNF-α in the serum and nasal mucosa lavage was measured using the LEGENDPlex^TM^ bead-based system for human cytokines kit (BioLegend, San Diego, CA, USA) following the manufacturer’s instructions. The concentration of each cytokines was calculated based on the respective standard curve (provided by the manufacture) that was obtained at the same time of the samples evaluation. The lower limit of detection for each cytokine was calculated by comparing to the reference value provided into the kit’s manufacturer over 2.5 standard deviations from these values. In relation to the cytokine concentrations obtained in the nasal mucosa lavage, the values initially found (pg/mL) were normalized by the total content of proteins, as measured by Bradford method [10], using the ratio of total protein (in milligrams) to cytokine concentrations in each sample.

### 2.6. Statistical Analysis

Continuous and semicontinuous data were initially compared with the Gaussian curve and normality was determined for each variable using Shapiro-Wilk test. Physical characteristics, aerobic capacity, conclusion time of the marathon and neutrophil infiltration were analyzed using Student’s *t*-test and are presented in mean and standard deviation (SD). Concentrations of antibacterial peptides and cytokines were analyzed using Kruskal–Wallis with Dunn post hoc test and are presented as the median with the respective quartiles. IL-10/IL-12p70 ratio was analyzed using a nonparametric *n*-sample signed rank test (Quade test) with equal medians of values from each time points as null hypotheses. In order to determine which sample inside the groups is different from the others, a post hoc Quade test was conducted with pairwise case analysis. A 2 × 2 contingency table McNemar Chi square test (with Yates correction) was used to determine whether the difference in the number of volunteers who presented URS, as well as in the duration of the symptoms between the two groups was significant. Differences at *p*-values < 0.05 were considered as statistically significant.

## 3. Results

As shown in Table 1, the groups´ stratification guaranteed the homogeneity of both volunteer´s group (*p* > 0.05), in terms of physical characteristics, aerobic capacity, and time of conclusion of the race.

### 3.1. Effects of LcS on the Incidence and Duration of Upper Respiratory Symptoms (URS)

Regarding the occurrence of URS or gastrointestinal disturbances, it is worthy clarifying that all volunteers were oriented to report any symptoms. However, in order to obtain the maximal information about the health status of all volunteers, we maintained a daily contact with them via telephone or social network. Before the marathon, no symptoms were reported. However, after the marathon ending, eight volunteers from group placebo (36.37%) reported URS, such as coryza, sneezing, itching, and burning sensation in the nose, as well as congested nose, dry throat, sore throat, cold, and influenza. In addition, the duration of the symptoms began in the first day and lasted up to 10 days post-marathon. From LcS group, only three volunteers (15%) reported URS, such as cough, congested nose, and dry nose with herpes lesion. The duration of the symptoms began in the first day and lasted up to 5 days post-marathon. Although these data suggests differences between the volunteer´s groups, the McNemar chi square statistical tests did not show significant differences both in the proportion (*p* = 0.076) and in the duration (*p* = 0.089) of upper airway symptoms between the volunteers of LcS and placebo groups.

### 3.2. LcS Maintains Secretory IgA (SIgA) and Antimicrobial Peptides Salivary Levels after the Marathon

As shown in Figure 3, immediately after the marathon, lower salivary levels of SIgA (Figure 3A) and defensing- 1 (Figure 3B) were observed in the placebo group as compared to values found in the time points Pre, 72 h, and 14 d and also in the LcS group post-marathon. Lower lysozyme (LZM, Figure 3C) and LL-37 (Figure 3D) levels were observed immediately after the marathon in the group placebo as compared to the value’s pre-marathon. Lactoferrin (LTF) levels (Figure 3E) remained unchanged in both groups.

### 3.3. LcS Reduces Neutrophils Infiltration in Nasal Mucosa

Table 2 shows that the neutrophils infiltration on the nasal mucosa Pre and 14 d after the marathon was similar between the groups, while a lower neutrophils infiltration were found in LcS group compared to placebo group in three time points: post-ingestion, immediately, and 72 h post-marathon.

### 3.4. LcS Modulates the Pro and Anti-inflammatory Cytokines Response in Upper Airways

Figure 4 shows that the concentrations of IL-1ra (Figure 4B), IL-4 (Figure 4C), and IL-12p70 (Figure 4G) in the nasal mucosa were unchanged in both groups. Higher IL-6 (Figure 4E) and IL-13 (Figure 4H) and TNF-α (Figure 4I) levels were observed immediately after the marathon in placebo group as compared to the values found in the other time points and also in the LcS group post-marathon. As for IL-5 (Figure 4D), the placebo group showed higher levels immediately after the marathon than the values found in the occasions Pre, 72 h, and 14 d and also in the LcS group post-marathon. Higher IL-1β levels (Figure 4A) were found in the placebo group immediately after the marathon compared to the values found in the time points Post-ingestion, 72 h and 14 d and in the LcS group post-marathon. IL-10 (Figure 4F) was found to be higher levels in the LcS group immediately after the marathon compared to the values observed in the other time points and in the placebo group post-marathon.

### 3.5. LcS Modulates the Pro- and Anti-Inflammatory Cytokines Response in Systemic Circulation

Figure 5 shows that serum concentration of IL-1β (Figure 5A), IL-4 (Figure 5C), IL-5 (Figure 5D), and IL-13 (Figure 5H) was similar in both groups. Higher IL-6 (Figure 5D), IL-10 (Figure 5F), and IL-12p70 (Figure 4G) levels were observed in both volunteer groups immediately after the marathon compared to the values obtained in the other time points. Whereas higher concentration of IL-1ra (Figure 5B) was found in the placebo group immediately after the marathon when compared to values observed in other time points; in the LcS group it was observed that IL-1ra increased immediately after the marathon in comparison to the values obtained in the time points Pre, 72 h, and 14 d. Serum concentration of TNF-α (Figure 5I) in the placebo was higher immediately after the marathon than the values found in the other time points and in the LcS group post-marathon

### 3.6. LcS Increases the Anti-inflammatory Ratio in Upper Airways

Table 3 shows that the ratio of the IL-10 and IL-12 (IL-10/IL-12p70) levels in the serum was maintained similar during the study, whereas in the nasal mucosa lavage the values found in LcS group immediately after the marathon was higher than the values observed in the other time points and in the placebo group post-marathon.

## 4. Discussion

In this study we showed that the daily ingestion of fermented milk containing 40 billion of LcS during 30 days before a marathon competition was able to (1) maintain the salivary levels of both SIgA and antimicrobial peptides; (2) increase the nasal IL-10 levels, a classical anti-inflammatory cytokine, which led to the higher nasal IL-10/IL-12p70 ratio in LcS group immediately post-marathon; (3) reduce the nasal levels of proinflammatory cytokines, such as IL-1, IL-5, IL-6, IL-13, and TNF-; and (4) decrease the nasal mucosal neutrophil infiltration, demonstrating an anti-inflammatory effect induced by LcS in the upper airways. In a similar way, LcS group had lower serum TNF-α levels compared to placebo group immediately post-marathon. Both groups showed higher serum IL-1ra, IL-6, and IL-10 levels immediately post-marathon, displaying a classical anti-inflammatory pathway modulated by physical exercise [3,11]. 

Despite IL-6 being traditionally classified as a proinflammatory cytokine, together with IL-4 and IL-10, it is involved in the Th2 immune response, particularly in the protection against parasites and stimulation of humoral immunity [3]. In addition, elevations of IL-6 can be found in a variety of severe insults, such as trauma, burns, hemorrhagic shock, sepsis, and ischemia-reperfusion injuries together with IL-1 and TNF-α [12,13]. However, in terms of physical exercise, plasma IL-6 levels can increase up to 100-fold during and after a session of exhaustive exercise [3] in response to reduction of muscle glycogen and also in response to contracting skeletal muscle fibers [14,15]. When released from skeletal muscle, IL-6 is considered as a myokine and it presents metabolic actions both locally by enhancing glucose uptake in skeletal muscle and systemically by inducing lipolysis in the adipose tissue, beyond to induce glycogenolysis in the hepatic tissue [15]. Moreover, IL-6 from muscle can also act in the regulation of the inflammation, since this cytokine precedes the release of anti-inflammatory cytokines such as IL-1ra and IL-10 [15,16,17]. Therefore, the increased serum concentration of IL-6, IL-1ra, and IL-10 found in our study in both volunteer groups was an expected result and it was not influenced by ingestion of test drink with or without LcS.

Although the literature highlights that systemic TNF-α levels remains unchanged after a marathon [3,17], probably due to the inhibitory effect of IL-6-induced IL-1ra and IL-10, elevations in the serum TNF-α levels can also be found after a prolonged endurance exercise [18], as observed in the placebo group of the present study. So, the daily ingestion of LcS was able to prevent the elevation of systemic TNF-α levels during an exhaustive exercise session, as run a marathon. Corroborating these findings, it was demonstrated that after 4, 10, and 14 weeks of LcS ingestion, CD14^+^ blood cells from healthy adults stimulated with LPS showed a reduction in the TNF-α release [19]. 

In a similar way, even though systemic IL-12p70 levels could remain unchanged post-marathon [17,20], it was observed increased plasma IL-12 levels after an anaerobic maximal cycle ergometer exercise [21], as found here in both groups immediately after the marathon. Since the IL-12p70 is a proinflammatory cytokine [21], together with the elevation of serum TNF-α levels observed in placebo group immediately after the marathon, it is possible to suggest that an acute proinflammatory status was induced by the race. Regarding the LcS group, it has been reported that LcS can stimulate different leukocytes (i.e., human PBMC, monocytes, and macrophages) to produce IL-12 [22,23], which, in agreement with Takeda and colleagues [23], is capable to augment NK cell activity. So, we hypothesized that the increased levels of serum IL-12p70 observed in the LcS group could enhance NK cell activity, and consequently mitigate the development of illness induced by the proinflammatory response after the marathon, as observed in this study.

As previously highlighted by our group, not only alterations in the systemic, but also in the upper airways immune/inflammatory responses are found after a strenuous and prolonged exercise session, as a marathon race [4]. Elevations of TNF-α and IL-1β—two well-known proinflammatory cytokines—can be verified both in acute and chronic airway inflammation [24,25]. Among some properties, both cytokines are responsible for the neutrophil recruitment in the nasal mucosa, since the combined treatment with anti-IL-1β and TNF-α was able to reduce or even abolish neutrophil infiltration in the nasal mucosa and lung tissue in a model of pathological allergic inflammatory reaction [25]. Therefore, the increased levels of these cytokines found in the nasal lavage obtained immediately after the marathon in the placebo group can putatively be involved in the elevation of neutrophil infiltration on nasal mucosa, as observed 72 h after the marathon in the placebo group. In addition, it was reported that both TNF-α and IL-1β precedes the IL-6 release by primary upper airway epithelial cells obtained of healthy volunteers [26].

Corroborating our former studies [4,27], the higher IL-6 levels found in the upper airways immediately after a marathon in the placebo group was a result expected and can be involved in an induction of a Th2 immune response. In addition, the increased levels of IL-5 and IL-13 found in the upper airways of the placebo group reinforce that a Th2 immune response was induced by the marathon. It is widely accepted that the upper airways inflammatory response induced by the marathon can be influenced by hyperventilation, in which leads to mucosal dryness and higher exposition to air pollutants. Ramanathan and colleagues [28] showed not only increased levels of IL-13 in the nasal lavage, but also higher eosinophils infiltration on respiratory airways of mice after exposition to pollutants. In relation to IL-5, it is well-known that this cytokine participates in both recruitment and activation of eosinophils and the presence of these immune cells in the upper airways also generate cytokines that may contribute to the perpetuation of Th2 inflammation [29]. 

On the other hand, the LcS group showed higher levels of IL-10 immediately after the marathon. The anti-inflammatory role of IL-10 in the airways has been proved over the past 10 years [30], and in relation to the marathon, our group previously demonstrated that an increased IL-10 level was found immediately after a marathon in a group of amateurs’ marathon runners who did not present URS post-race. So, in that study, we postulated that higher IL-10 levels help improve the protection of the upper respiratory tract against mucosal inflammation induced by the marathon [4]. Regarding LcS actions, it was observed that LcS was able to stimulate the immune cells, especially macrophages and dendritic cells, to secrete different cytokines as IL-10, IL-12, IFN-, and TNF-α [31,32,33]. Based on the fact that IL-10 and IL-12 exhibits opposing roles in the regulation of the immune responses, the IL-10/IL-12 ratio has been used as a crucial measure of the balance between the anti-inflammatory and proinflammatory states induced by LcS [34]. So, our remarkable finding of an expressive increase in the IL-10/IL-12 ratio in the upper airway mucosa of the LcS group immediately post-marathon demonstrates that the daily intake of a fermented milk containing LcS was able to induce an anti-inflammatory response that can mitigate the deleterious effects of marathon on the mucosal inflammation.

Secretory immunoglobulin A (SIgA) is considered as the “first line of defense” against mucosal pathogens, and there is a consensus that reduced salivary SIgA levels are associated with an increased risk of developing URS after an exhaustive physical exercise session or even during periods of intense exercise training [3]. Here, lower salivary SIgA levels were found immediately post-marathon only in the placebo group. As previously reported [8], the daily ingestion of LcS favors the increase of salivary SIgA levels in healthy adult individuals. Although we did not observe increase in SIgA in LcS group, the maintenance of it levels can contribute to the upper airways protection against pathogens and consequently minimizing the incidence and duration of URS, as already mentioned [3].

Even though the majority of the researches considers salivary SIgA as the main factor of mucosal immunity and agrees that its levels could favor or not the incidence of URS, the importance of other mucosal immunological agents, as the antimicrobial peptides, has been recognized. These peptides are molecules with a low molecular mass that exhibit preeminent inhibitory activity against bacteria, viruses and fungi, being considered as essential elements of the innate immune system [35]. In response to an exhaustive physical exercise session, it was demonstrated that elite swimmers showed a significant reduction in the salivary lysozyme after an intense exercise training session [36,37]. Furthermore, lower salivary levels of lactoferrin and lysozyme were found in a group of elite rower athletes during a competition period as compared to sedentary individuals [38]. In a similar way to that described for SIgA, reductions in the salivary levels of defesin- 1, LL-37 and lysozyme were also observed immediately after the marathon in the placebo group, but not in LcS group. So, these remarkable results showed that the daily ingestion of fermented milk containing LcS can maintain the immune protection in the upper airway mucosal and minimize both the incidence and duration of URS after a marathon, as observed by us.

## 5. Conclusion

Taken together, our results showed, for the first time, that the previous 30 days daily ingestion of fermented milk containing 40 billion of LcS was able to modulate both immunological and inflammatory responses in the blood and also in the upper airways mucosal of amateurs´ runners after a marathon, presenting protective effects.

## Figures and Tables

**Figure 1 nutrients-11-01678-f001:**
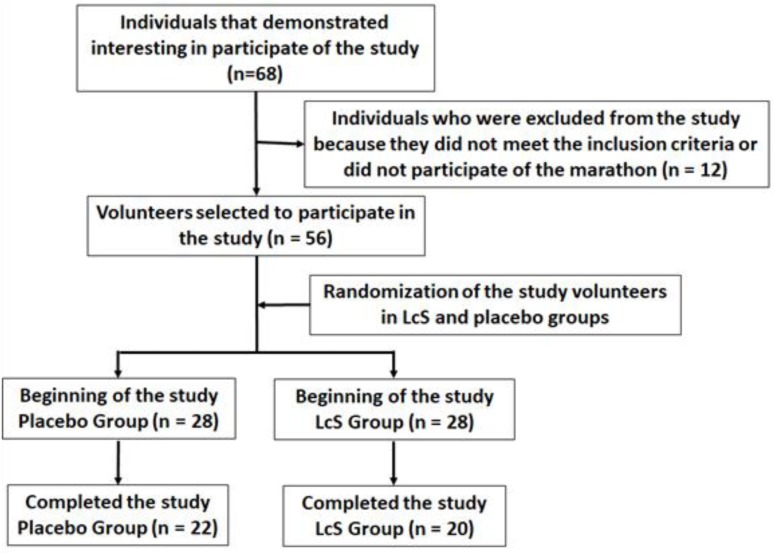
Flow diagram of the study.

**Figure 2 nutrients-11-01678-f002:**
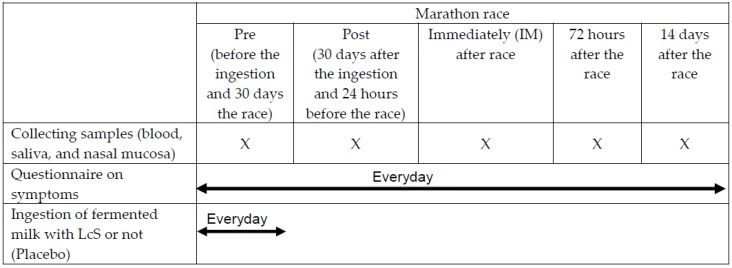
Experimental design.

**Figure 3 nutrients-11-01678-f003:**
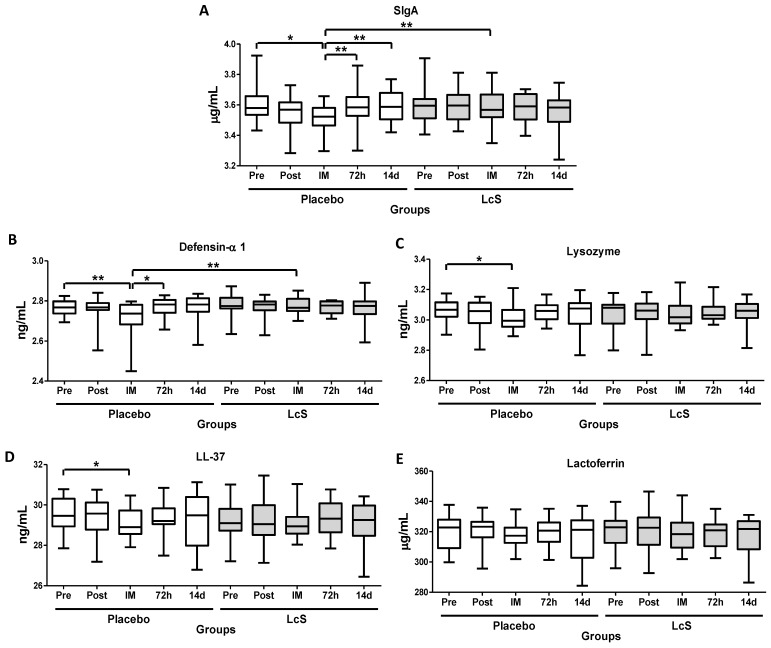
LcS maintain salivary immune protection after the marathon. Salivary concentration of secretory immunoglobulin A (SIgA, g/mL, (**A**) and the antimicrobial peptides defensing-1 (ng/mL (**B**), lysozyme (ng/mL (**C**), cathelicidin LL-37 (ng/mL (**D**), and lactoferrin (g/mL (**E**) in the volunteers of the placebo and LcS groups at five different occasions: before (Pre) and 30 days after the ingestion of the fermented milk containing or not containing LcS (Post-ingestion); immediately (IM); 72 hours (72 h) and 14 days (14 d) after the marathon ends. Values are presented in the median with the respective quartiles. The risk value was set at 5% (*p* < 0.05). * *p* < 0.05 and ** *p* < 0.01.

**Figure 4 nutrients-11-01678-f004:**
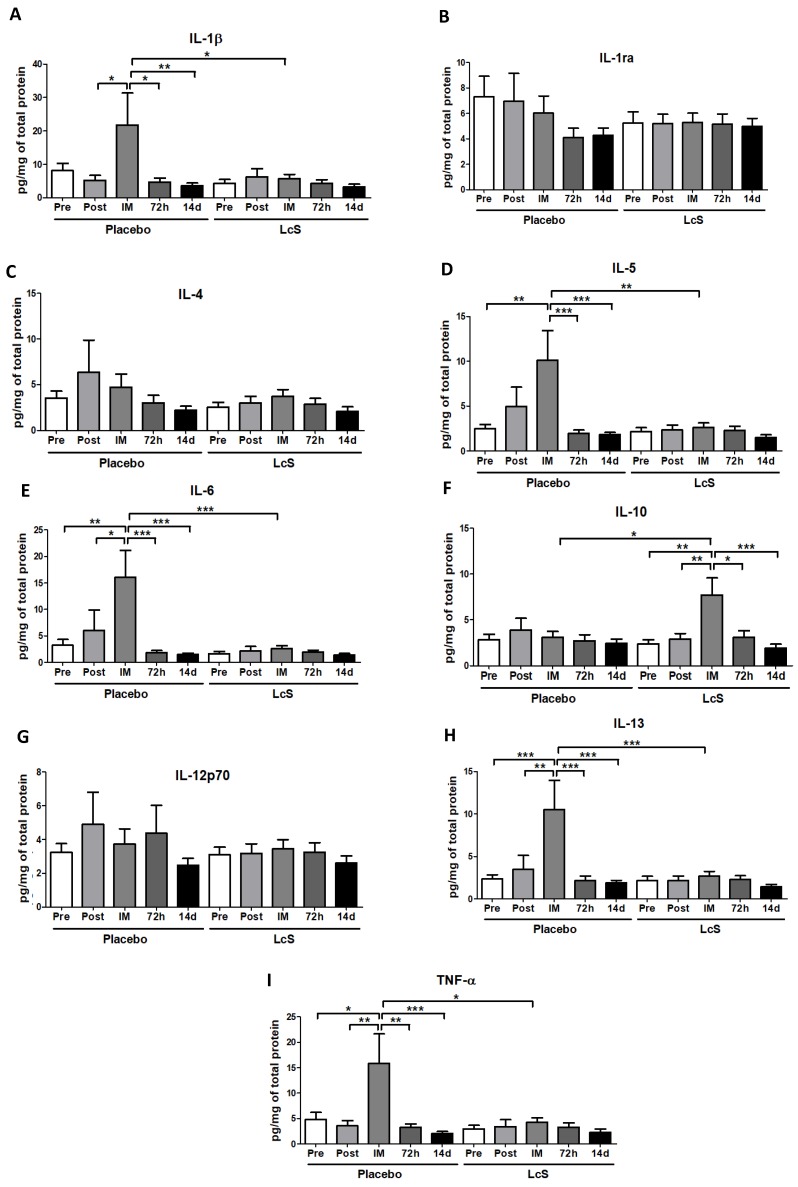
LcS increase anti-inflammatory response on the upper airways after the marathon. Concentration (pg/mg of total protein) of IL-1β (**A**)**,** IL-1ra (**B**)**,** IL-4 (**C**)**,** IL-5 (**D**)**,** IL-6 (**E**)**,** IL-10 (**F**), IL-12p70 (**G**)**,** IL-13 (**H**), and TNF- (**I**) in nasal mucosal lavage of the volunteers in the placebo and LcS groups at five different occasions: before (Pre) and 30 days after the ingestion of the fermented milk containing or not containing LcS (Post-ingestion); immediately (IM); 72 hours (72 h) and 14 days (14 d) after the marathon ends. Values are presented in median with the respective quartiles. * *p* < 0.05; ** *p* < 0.01 and *** *p* < 0.001.

**Figure 5 nutrients-11-01678-f005:**
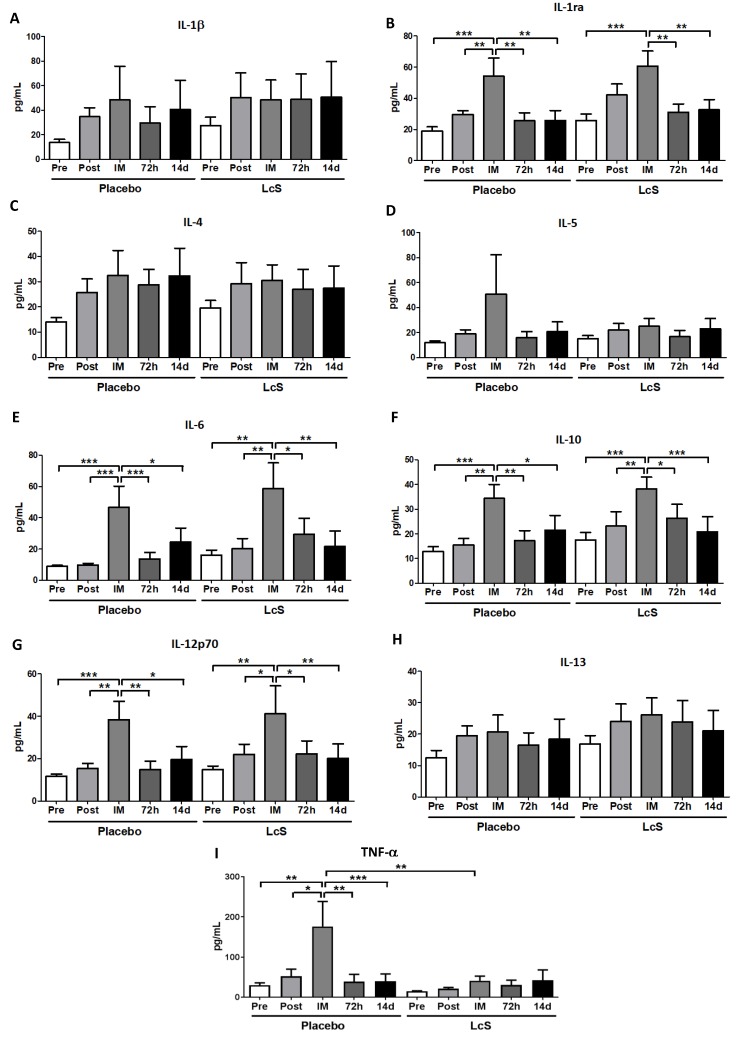
LcS increase systemic IL-12p70 levels after the marathon Serum concentration (pg/mL) of IL-1β (**A**)**,** IL-1ra (**B**)**,** IL-4 (**C**)**,** IL-5 (**D**)**,** IL-6 (**E**), IL-10 (**F**)**,** IL-12p70 (**G**), IL-13 (**H**), and TNF- (**I**) in the placebo and LcS groups at five different occasions: before (Pre) and 30 days after the ingestion of the fermented milk containing or not containing LcS (Post-ingestion); immediately (IM); 72 hours (72 h) and 14 days (14 d) after the marathon. Values are presented in median with the respective quartiles. * *p* < 0.05; ** *p* < 0.01 and *** *p* < 0.001.

**Table 1 nutrients-11-01678-t001:** Physical characteristics (age, weight, height, total body fat, total muscle mass and body mass index (BMI)), and aerobic capacity (relative VO_2_max) of the volunteers allocated in the groups LcS and placebo. All data were analyzed using Student’s t-test and are presented in mean and standard deviation. The level of significance was 5% (*p* < 0.05).

Variables	Volunteers (*n* = 42)	
	Placebo (*n* = 22)	LcS (*n* = 20)
Age (year)	40.1 ± 10.3	39.6 ± 8.8
Weight (kg)	76.5 ± 10.4	72.4 ± 7.8
Height (cm)	176.9 ± 7.6	173.8 ± 6.4
Total body fat (%)	18.6 ± 7.5	16.5 ± 6.6
Total muscle mass (g)	58.1 ± 4.7	54.7 ± 5.9
BMI (kg/m^2^)	24.4 ± 2.2	23.4 ± 2.4
Relative VO_2_max (ml/kg/min)	57.64 ± 6.89	57.86 ± 6.85
Time of conclusion of the race (hours)	04:09:20 ± 00:19:56	04:15:30 ± 00:15:22

**Table 2 nutrients-11-01678-t002:** Percentage of neutrophil infiltration on nasal mucosa of the study volunteers separated into groups LcS and placebo. All data are presented in mean and standard deviation and were analyzed by Student’s t-test. The level of significance was 5% (*p* < 0.05).

Variable	Volunteers (*n* = 42)					*p* Value
	Placebo (*n* = 22)					
	Pre	Post	IM	72 h	14 d	
Neutrophil (%)	50 ± 3	43 ± 6 *	38 ± 3 *^,#^	53 ± 4 *	56 ± 3	<0.05
	LcS (*n* = 20)					
	45 ± 4 ^ø^	21 ± 3	28 ± 2	29 ± 3	48 ± 6 ^ø^	<0.05

* differences between placebo and LcS group for the same occasion; ^#^ differences between IM and Pre, 72 h and 14 d in placebo group; ^ø^ differences between Pre or 14 d and Post, and IM and 72 h in LcS group.

**Table 3 nutrients-11-01678-t003:** Ratio of IL-10 and IL-12p70 (IL-10/IL-12p70) in the serum and nasal mucosa lavage of the study volunteers separated into groups LcS and placebo obtained in five different occasions (Pre, Post, IM. 72 h e 14 d). All data are presented in mean and standard deviation and were analyzed by a nonparametric *n*-sample signed rank test (Quade test) with a post hoc Quade testing was conducted with pairwise case analysis. The level of significance was 5% (*p* < 0.05).

	Variable/Occasion	IL-10/IL-12p70 ratio				
Group/Sample		Pre	Post	IM	72 h	14 d
Placebo/Serum		1.05 ± 0.10	1.12 ± 0.16	1.63 ± 0.43	1.28 ± 0.19	1.16 ± 0.11
LcS/Serum		1.09 ± 0.11	1.07 ± 0.13	1.40 ± 0.15	1.30 ± 0.13	1.11 ± 0.09
Placebo/Nasal mucosa lavage		1.20 ± 0.47	0.83 ± 0.10	1.12 ± 0.13	0.84 ± 0.11	1.00 ± 0.11
LcS/Nasal mucosa lavage		0.81 ± 0.10	0.94 ± 0.13	5.84 ± 2.74 *^,#^	0.95 ± 0.12	0.76 ± 0.10

* differences between placebo and LcS group for the same occasion (*p* < 0.001); ^#^ differences between IM (immediately post-marathon) and the other occasions in the LcS group (*p* < 0.001).

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
