# Peer review of "Daily Intake of Fermented Milk Containing Lactobacillus casei Shirota (Lcs) Modulates Systemic and Upper Airways Immune/Inflammatory Responses in Marathon Runners"

_nutrients, 2019, doi:10.3390/nu11071678_

Round 1

Reviewer 1 Report

Vaisberg et colleagues described the effect of fermented milk containing Lactobacillus casei on upper airway immune response of marathon runners. The authors reported the analysis of pro- and anti-inflammatory cytokines in nasal secretion and serum of athletes drinking fermented milk or placebo before and after a marathon. The manuscript is well structured and clear. For each topic, a detailed introduction and discussion were described. The work is very interesting but there are some points that need to be improved.

Major comments:

The placebo used was milk not fermented, to affirm the immunostimulatory role of L. casei the correct placebo could be milk fermented with probiotics with low or no effect on immune system. The authors should justify the choice of unfermented milk.

Minor comments

Abstract: the composition of placebo should be indicated

Page 4 line 105: the time of fermentation should be indicated. Metabolites produced by probiotics could play an important role in the biological activity. The incubation time of fermentation is a good point for the metabolites production.

Page 4, lines 116-118: For the description of the samples collection, a flow chart could have a better impact

Page 4, lines 139-147: Paragraphs of SigA and antimicrobial peptides determination should be merged.

Page 5, table 1: the P value (not significant) could be deleted, the description of the result in the test is enough.

Page 6, figure 2: The title of histograms should be indicated. For a better reading of the graphs, the corresponding bars (Placebo and LcS groups) should be indicated (different colour or different filling).

Page 7, table 2: data of placebo and LcS groups should be placed on two lines to favour the comparison for the sampling times. Moreover, the differences observed inside the placebo or LcS groups should be described in the text.

Page 12, last line: Reference 31 is a paper of Shida of 2006, the recent papers described the production of cytokines by macrophages stimulated with L. casei should be mentioned:

Zielińska D, Długosz E, Zawistowska-Deniziak A. Functional Properties of Food Origin Lactobacillus in the Gastrointestinal Ecosystem-In Vitro Study. Probiotics Antimicrob Proteins. 2018 Aug 23. doi: 10.1007/s12602-018-9458-z. [Epub ahead of print]

De Marco S, Sichetti M, Muradyan D, et al. Probiotic Cell-Free Supernatants Exhibited Anti-Inflammatory and Antioxidant Activity on Human Gut Epithelial Cells and Macrophages Stimulated with LPS. Evid Based Complement Alternat Med. 2018 Jul 4;2018:1756308. doi: 10.1155/2018/1756308. eCollection 2018.

Author Response

Comments and Suggestions for Authors
Vaisberg et colleagues described the effect of fermented milk containing Lactobacillus casei on upper airway immune response of marathon runners. The authors reported the analysis of pro- and anti-inflammatory cytokines in nasal secretion and serum of athletes drinking fermented milk or placebo before and after a marathon. The manuscript is well structured and clear. For each topic, a detailed introduction and discussion were described. The work is very interesting but there are some points that need to be improved.
Major comments:
Reviewer: The placebo used was milk not fermented, to affirm the immunostimulatory role of L. casei the correct placebo could be milk fermented with probiotics with low or no effect on immune system. The authors should justify the choice of unfermented milk.
Authors’ answer: Thank you very much for your review of our manuscript, which help us to improve it. Unfortunately, the company was able to provide us only an unfermented milk as placebo, since the fermentation occurs in the presence of the probiotics. By the way, for sure, for further studies there is a need to use another fermented milk as a placebo, perhaps using another type of innocuous probiotic. However, we would like to clarify that the scientific literature presents several studies aiming to elucidate the effects of LcS in the modulation of immune response in athletes or non-athletes, in which the authors reported to use unfermented milk as placebo. Below some studies using unfermented milk as a placebo are listed.
1- Tamura M, Shikina T, Morihana T, Hayama M, Kajimoto O, Sakamoto A, Kajimoto Y, Watanabe O, Nonaka C, Shida K, Nanno M. Effects of probiotics on allergic rhinitis induced by Japanese cedar pollen: randomized double-blind, placebo-controlled clinical trial. Int Arch Allergy Immunol. 2007;143(1):75-82.
2- Gleeson M, Bishop NC, Oliveira M, Tauler P. Daily probiotic's (Lactobacillus
casei Shirota) reduction of infection incidence in athletes. Int J Sport Nutr Exerc
Metab. 2011 Feb;21(1):55-64.
3- Gleeson M, Bishop NC, Struszczak L. Effects of Lactobacillus casei Shirota
ingestion on common cold infection and herpes virus antibodies in endurance
athletes: a placebo-controlled, randomized trial. Eur J Appl Physiol. 2016
Aug;116(8):1555-63. doi: 10.1007/s00421-016-3415-x.
4- Kato-Kataoka A, Nishida K, Takada M, Kawai M, Kikuchi-Hayakawa H, Suda
K, Ishikawa H, Gondo Y, Shimizu K, Matsuki T, Kushiro A, Hoshi R, Watanabe
O, Igarashi T, Miyazaki K, Kuwano Y, Rokutan K.
Fermented Milk Containing Lactobacillus casei Strain Shirota Preserves the
Diversity of the Gut Microbiota and Relieves Abdominal Dysfunction in Healthy
Medical Students Exposed to Academic Stress. Appl Environ Microbiol. 2016
May 31;82(12):3649-58. doi: 10.1128/AEM.04134-15.
5- Shida K, Sato T, Iizuka R, Hoshi R, Watanabe O, Igarashi T, Miyazaki
K, Nanno M, Ishikawa F. Daily intake of fermented milk with Lactobacillus
casei strain Shirota reduces the incidence and duration of upper respiratory
tract infections in healthy middle-aged office workers. Eur J Nutr. 2017
Feb;56(1):45-53. doi: 10.1007/s00394-015-1056-1.
6- Lei M, Guo C, Wang D, Zhang C, Hua L. The effect of probiotic Lactobacillus
casei Shirota on knee osteoarthritis: a randomised double-blind, placebocontrolled
clinical trial. Benef Microbes. 2017 Oct 13;8(5):697-703. doi:
10.3920/BM2016.0207.
Minor comments
Reviewer: Abstract: the composition of placebo should be indicated.
Authors’ answer: As required, we have indicated “unfermented milk” in the
placebo description. In addition, we would like to clarify that in accordance with
the official response given by the Yakult Company to us: "the company cannot
disclose the detailed recipe of the placebo due to our company policy. All we
can say about the placebo is that the unfermented milk (placebo) was basically
prepared with the same ingredients of test product but without LcS”.
Reviewer: Page 4 line 105: the time of fermentation should be indicated.
Metabolites produced by probiotics could play an important role in the biological
activity. The incubation time of fermentation is a good point for the metabolites
production.
Authors’ answer: Thank you very much for the information concerning this
subject. However, similarly to the mentioned in the response above, the official
response given by the Yakult Company was "we cannot disclose the
fermentation time of the product because the manufacturing condition is a toplevel
company secret”.
Reviewer: Page 4, lines 116-118: For the description of the samples collection,
a flow chart could have a better impact.
Authors’ answer: Thank you so much for this recommendation and the figure
below showing the description of the Experimental Design was inserted in the
manuscript.

(Please see the attachment.)

Reviewer: Page 4, lines 139-147: Paragraphs of SIgA and antimicrobial
peptides determination should be merged.
Authors’ answer: Thank you so much for the recommendation and the
description of SIgA and antimicrobial peptides determination were merged.
Reviewer: Page 5, table 1: the P value (not significant) could be deleted, the
description of the result in the test is enough.
Authors’ answer: Thank you for the comment and we removed the P value
column of the table.
Reviewer: Page 6, figure 2: The title of histograms should be indicated. For a
better reading of the graphs, the corresponding bars (Placebo and LcS groups)
should be indicated (different colour or different filling).
Authors’ answer: Thank you so much for the recommendation. We inserted
the title of histograms and also changed the graphs indicating, with different
filling, the bars corresponding to the Placebo and LcS groups. In addition, we
also inserted the information “Values are presented in the median with the
respective quartiles” in the figure´s legend.

(Please see the attachment.)

Reviewer: Page 7, table 2: data of placebo and LcS groups should be placed
on two lines to favour the comparison for the sampling times. Moreover, the
differences observed inside the placebo or LcS groups should be described in
the text.
Authors’ answer: Thank you so much for the recommendation. We changed
the table to favour the comparison between the data obtained for Placebo and
LcS groups and for the sampling times. It is worth to clarify that the table 2 was
changed to table 2. (Please see the attachment.)

Marco S, Sichetti M, Muradyan D, et al. Probiotic Cell-Free Supernatants
Exhibited Anti-Inflammatory and Antioxidant Activity on Human Gut Epithelial
Cells and Macrophages Stimulated with LPS. Evid Based Complement Alternat
Med. 2018 Jul 4;2018:1756308. doi: 10.1155/2018/1756308. eCollection 2018.
Authors’ answer: Thank you so much for the recommendation. We inserted
these references into the manuscript as requested.

Reviewer 2 Report

General Comments: The manuscript addresses critical questions in the field of nutrition relevant to the potential short term effects of dietary intake of probiotics on physical performance and provide new information that may indicate a modifying effect on immune status.  The investigators have studied the effect of fermented milk containing Lactobacillus casei Shirota (LcS) in marathon runners and have evaluated immune responses in nasal mucosa and saliva, as a reflection of upper airways’ immune status, and in serum from systemic circulation.  The sample size of evaluable subjects receiving LcS was slightly lower than those who drank the placebo for a month before the marathon.  The analysis included pairwise comparisons to assess significance.  Strengths of the study include the design, specifically the use of multiple measures of immune response at 3 time points before and after the marathon and the strong choice of parameters, a panel of inflammatory cytokines for serum and nasal lavage, secretory IgA (SIgA) and antimicrobial peptides in saliva, and neutrophil infiltration as a functional determinant.  The inclusion criteria selected for runners with a history of upper respiratory symptoms after physical exhaustion. The data showed similarity in the group receiving LcS compared to the placebo group including racing time.  Differences in symptoms post marathon were observed that could not be confirmed by statistical analysis while selected parameters of immune reactivity were significantly affected by LcS post marathon (SIgA, defensin alpha-1, lysozyme, and cathelicidin compared to placebo while lactoferrin was not.  The error bars indicate wide variation in the groups for all parameters in Figure 2.  In contrast the data for neutrophil infiltration showed a parallel effect with low standard deviation from the mean as did the cytokine data. The ratio data are especially pertinent in this study. Therefore the conclusions appear to be well supported by the data.

Questions and Comments: 

     1.  Although this was a double blind randomized clinical trial, it seems likely that the fermented milk would be recognizable from unfermented milk. If so the implications may affect confidence and attitudes of the participant runners and perhaps runner’s perception of symptoms and should be discussed.

     2. The levels of cytokines present in nasal lavage vary widely according to which cytokine is being measured. Figure 3 would be more illuminating if the y-axis was standardized for pg/mg total protein. Clearly this difference also affects detection of significance and biological impact.

     3. The above comment is also true for Figure 4, but to a lesser extent since no cytokines are present in amounts less than 10pg/ml across all groups as seen for IL-1ra and IL-12p70 in Figure 3.

Author Response

Comments and Suggestions for Authors
General Comments: The manuscript addresses critical questions in the field of nutrition relevant to the potential short term effects of dietary intake of probiotics on physical performance and provide new information that may indicate a modifying effect on immune status. The investigators have studied the effect of fermented milk containing Lactobacillus casei Shirota (LcS) in marathon runners and have evaluated immune responses in nasal mucosa and saliva, as a reflection of upper airways’ immune status, and in serum from systemic circulation. The sample size of evaluable subjects receiving LcS was slightly lower than those who drank the placebo for a month before the marathon. The analysis included pairwise comparisons to assess significance. Strengths of the study include the design, specifically the use of multiple measures of immune response at 3 time points before and after the marathon and the strong choice of parameters, a panel of inflammatory cytokines for serum and nasal lavage, secretory IgA (SIgA) and antimicrobial peptides in saliva, and neutrophil infiltration as a functional determinant. The inclusion criteria selected for runners with a history of upper respiratory symptoms after physical exhaustion. The data showed similarity in the group receiving LcS compared to the placebo group including racing time. Differences in symptoms post marathon were observed that could not be confirmed by statistical analysis while selected parameters of immune reactivity were significantly affected by LcS post marathon (SIgA, defensin alpha-1, lysozyme, and cathelicidin compared to placebo while lactoferrin was not. The error bars indicate wide variation in the groups for all parameters in Figure 2. In contrast the data for neutrophil infiltration showed a parallel effect with low standard deviation from the mean as did the cytokine data. The ratio data are especially pertinent in this study. Therefore the conclusions appear to be well supported by the data.
Questions and Comments:
Reviewer: 1. Although this was a double-blind randomized clinical trial, it seems likely that the fermented milk would be recognizable from unfermented
milk. If so the implications may affect confidence and attitudes of the participant
runners and perhaps runner’s perception of symptoms and should be
discussed.
Authors’ answer: Thank you very much for your review of our manuscript,
helping us to improve it significantly. Concerning your comment, we would like
to clarify that the use of an unfermented milk as placebo was based on several
studies that aimed to elucidate the effect of LcS in the modulation of immune
response in athletes or non-athletes. In all the studies listed below, the
unfermented milk was used as placebo. In addition, it is worth to mention that
during all the study period none of the volunteers requested information´s about
the compounds presents into the beverage ingested or reported any problem
concerning the taste, color, and pH, that could lead to the recognition of
unfermented beverage as a placebo. In addition, corroborating the use of
unfermented milk as a placebo, as we used by us, some studies are listed
below:
1- Tamura M, Shikina T, Morihana T, Hayama M, Kajimoto O, Sakamoto
A, Kajimoto Y, Watanabe O, Nonaka C, Shida K, Nanno M. Effects of probiotics
on allergic rhinitis induced by Japanese cedar pollen: randomized double-blind,
placebo-controlled clinical trial. Int Arch Allergy Immunol. 2007;143(1):75-82.
2- Gleeson M, Bishop NC, Oliveira M, Tauler P. Daily probiotic's (Lactobacillus
casei Shirota) reduction of infection incidence in athletes. Int J Sport Nutr Exerc
Metab. 2011 Feb;21(1):55-64.
3- Gleeson M, Bishop NC, Struszczak L. Effects of Lactobacillus casei Shirota
ingestion on common cold infection and herpes virus antibodies in endurance
athletes: a placebo-controlled, randomized trial. Eur J Appl Physiol. 2016
Aug;116(8):1555-63. doi: 10.1007/s00421-016-3415-x.
4- Kato-Kataoka A, Nishida K, Takada M, Kawai M, Kikuchi-Hayakawa H, Suda
K, Ishikawa H, Gondo Y, Shimizu K, Matsuki T, Kushiro A, Hoshi R, Watanabe
O, Igarashi T, Miyazaki K, Kuwano Y, Rokutan K.
Fermented Milk Containing Lactobacillus casei Strain Shirota Preserves the
Diversity of the Gut Microbiota and Relieves Abdominal Dysfunction in Healthy
Medical Students Exposed to Academic Stress. Appl Environ Microbiol. 2016
May 31;82(12):3649-58. doi: 10.1128/AEM.04134-15.
5- Shida K, Sato T, Iizuka R, Hoshi R, Watanabe O, Igarashi T, Miyazaki
K, Nanno M, Ishikawa F. Daily intake of fermented milk with Lactobacillus
casei strain Shirota reduces the incidence and duration of upper respiratory
tract infections in healthy middle-aged office workers. Eur J Nutr. 2017
Feb;56(1):45-53. doi: 10.1007/s00394-015-1056-1.
6- Lei M, Guo C, Wang D, Zhang C, Hua L. The effect of probiotic Lactobacillus
casei Shirota on knee osteoarthritis: a randomised double-blind, placebocontrolled
clinical trial. Benef Microbes. 2017 Oct 13;8(5):697-703. doi:
10.3920/BM2016.0207.
Reviewer: 2. The levels of cytokines present in nasal lavage vary widely
according to which cytokine is being measured. Figure 3 would be more
illuminating if the y-axis was standardized for pg/mg total protein. Clearly this
difference also affects detection of significance and biological impact.
Authors’ answer: Thank you very much for your revision of our manuscript,
which help us to improve it significantly. We would like to clarify that the
concentration of each cytokines was calculated based on the respective
standard curve (provided by the manufacture) that was obtained at the same
time of the samples evaluation. The lower limit of detection for each cytokine
was calculated by comparing to the reference value provided into the kit's
manufacturer over 2.5 standard deviations from these values. In relation to the
cytokine concentrations obtained in the nasal mucosa lavage, the values initially
found (pg/mL) were normalized by the total content of proteins, as measured by
Bradford method [10], using the ratio of total protein (in milligrams) to cytokine
concentrations in each sample. Also, the legends of Figures 2 and 3 were rewritten
to better describe the graphical data. These information´s were added to
the manuscript text.
Reviewer: 3. The above comment is also true for Figure 4, but to a lesser
extent since no cytokines are present in amounts less than 10pg/ml across all
groups as seen for IL-1ra and IL-12p70 in Figure 3.
Authors’ answer: Thank you so much for your comment and the response to this question was given as the same for the question above (question 2).
